# BIRD: Generalizable Backdoor Detection and Removal for Deep Reinforcement Learning

**Xuan Chen[1], Wenbo Guo[1, 2], Guanhong Tao[1], Xiangyu Zhang[1], Dawn Song[2]**

[1]Purdue University
[2]UC Berkeley
{chen4124, henrygwb, taog, xyzhang}@cs.purdue.edu
{henrygwb, dawnsong}@berkeley.edu

## Abstract

Backdoor attacks pose a severe threat to the supply chain management of deep reinforcement learning (DRL) policies. Despite initial defenses proposed in recent studies, these methods have very limited generalizability and scalability. To address this issue, we propose BIRD, a technique to detect and remove backdoors from a pretrained DRL policy in a clean environment without requiring any knowledge about the attack specifications and accessing its training process. By analyzing the unique properties and behaviors of backdoor attacks, we formulate trigger restoration as an optimization problem and design a novel metric to detect backdoored policies. We also design a finetuning method to remove the backdoor, while maintaining the agent's performance in the clean environment. We evaluate BIRD against three backdoor attacks in ten different single-agent or multi-agent environments. Our results verify the effectiveness, efficiency, and generalizability of BIRD, as well as its robustness to different attack variations and adaptions.

## 1 Introduction

Recent studies [59, 20, 49, 8] demonstrate that an attacker can inject a backdoor into a DRL agent's policy. In particular, the attacker adds a trigger into the environmental state during the training process and forces the agent to take a backdoored/poisoned action that diminishes its total reward given by the environment. To accomplish this, they manipulate the agent's reward function and assign a high reward whenever the agent takes a poisoned action at a poisoned state. We denote the original reward given by the environment as the *actual reward* and the attacker-manipulated reward function as the *poisoned reward function*. As a result, the agent learns to take poisoned actions at these states, expecting the environment to assign a high actual reward to it. However, taking the poisoned actions actually diminishes the agent's actual reward and even causes the agent to fail its corresponding tasks.

Given that a backdoored agent performs normally at clean states, it is challenging to detect and remove the backdoor from *a pretrained policy without accessing the training process, having knowledge about the attack specifications (i.e., trigger, poisoned actions, poisoned reward function), and accessing the poisoned states during testing.* Most existing backdoor defenses do not consider the sequential decision-making nature of DRL policies [46, 48, 16, 28, 13, 9, 25, 61, 30, 42, 51, 52] or assume access to the agent training process [11, 66, 56, 66]. Only a few methods provide defense for pretrained policies [2, 15]. However, they have several limitations that hinder their practicality. Specifically, the method proposed in [2] can only be applied to attacks with a perturbation patch as the trigger and cannot detect whether a policy is backdoored or not. Similarly, the defense proposed in [15] is only applicable to attacks where the trigger is an adversarial agent's specific actions.

We propose a generalizable backdoor detection and removal method for DRL policies. Given a pretrained agent's policy network and value function, along with the corresponding clean environment,

37th Conference on Neural Information Processing Systems (NeurIPS 2023).

we first formulate trigger restoration as an optimization problem. The objective function is crafted as searching for a perturbation to the state representation that forces the agent to take actions that maximize its value function. Since a backdoored agent is trained using a poisoned reward function that assigns high values when the trigger is present, maximizing its value function should identify the trigger. Second, we design a novel metric to detect whether the agent is backdoored or not using the resolved perturbation (denoted as restored trigger). As we will detail in Section 3.3, adding the restored trigger to the clean environment will cause the backdoored agent to receive a much lower actual reward from the environment but increase or not affect the clean agent's actual reward. Based on this behavior difference, we design our metric as the actual reward difference of the agent before and after adding the restored trigger to the environment. Finally, we propose a novel finetuning method to remove the detected backdoor. We design additional regularization terms to the finetuning objective function to maintain the finetuned agent's actual reward in the clean environment. We also introduce a neuron re-initialization mechanism to ensure the backdoor can be successfully removed even when the restored trigger is not exactly the same as the ground truth one in shape and size.

We denote our method as BIRD (**B**ackdoor **I**dentification and **R**emoval for **D**RL). We extensively evaluate BIRD in ten single-agent or multi-agent RL environments against three prevalent backdoor attacks. First, our results demonstrate that BIRD outperforms existing backdoor detection methods designed for supervised learning in detecting backdoored policies. Moreover, we show the superiority of BIRD over existing methods in removing the backdoor while maintaining the agent's actual reward in the original clean environment. Second, we verify the effectiveness of our key design choices through ablation studies, especially our backdoor detection metric. We demonstrate that compared with the widely used trigger size metric, our metric enables much higher detection accuracy *without requiring subtly tuning the threshold*. Third, we demonstrate the computational efficiency of BIRD and its insensitivity to hyper-parameter choices. Furthermore, we verify the robustness of BIRD against different variations in attack and two possible adaptive attacks. To the best of our knowledge, BIRD is the first DRL backdoor defense that can *detect and remove* backdoors in a clean environment without requiring access to the attack specifications and policy training process. Furthermore, BIRD is applicable to all kinds of existing attacks, making it a highly generalizable defense mechanism.

## 2 Related Work

**Attacks.** Recent research works propose backdoor attacks against single-agent DRL [59, 20, 50, 8]. These attacks, denoted as perturbation-based attacks, add a small perturbation patch to the victim agent's state as the trigger. A backdoored policy performs normally in a clean environment but takes poisoned action at poisoned states, leading to task failure. A follow-up work considers a two-agent setup with an adversarial agent and a victim agent [49]. Rather than perturbing the states, they leverage the adversarial agent's certain actions as the backdoor trigger (denoted as adversarial agent attack). More recent works generalize both perturbation-based attacks and adversarial agent attacks to multi-agent cooperative RL with a team of victim agents [8, 7].

**Detection and Defenses.** As mentioned in Section 1, only two existing defenses are developed to mitigate backdoors for a pretrained DRL policy. One method, proposed by [2], uses a trigger filter technique that maps possibly poisoned states back to their corresponding clean states. This method is designed for perturbation-based attacks and cannot be applied to adversarial agent attacks. Additionally, it relies on singular value decomposition, which limits its scalability in complicated RL environments with high-dimensional state spaces. Another method [15] also has limited generalizability in that it is only applicable to adversarial agent attacks.

Some other research works also extend adversarial sample attacks [4, 14] to DRL and propose testing-phase perturbation attacks [17, 32, 36, 63, 26, 40, 19], which perturb the environment at certain states to fail a pretrained agent. To defend against such attacks, researchers either generalize existing adversarial defenses designed for supervised learning to DRL [63, 62, 31, 55, 22, 57, 56] or model the attack and defense as a two-player game and train a robust agent to find the Nash equilibrium point [33, 60, 64]. Similarly, some robust RL methods also propose robust training techniques against random perturbations to the agent's observations [65, 66], actions [33, 44], or reward functions [67]. Due to differences in attack setups, these methods cannot be directly applied to our problem.

# 3 Methodology

## 3.1 Problem Setup

**Formulation.** Consider a Markov Decision Process (MDP) $\mathcal{M} = (\mathcal{S}, \mathcal{A}, P, R, \gamma)$, where $\mathcal{S}$ and $\mathcal{A}$ are the continuous or discrete state and action space. $P$ is the state transition function, $R$ is the reward function, and $\gamma$ is the discount factor. Without facing any attack, the agent's goal is to learn an optimal policy $\pi$ that maximizes its expected long-term reward, $\eta(\pi) = \mathbb{E}[\sum_t \gamma^t R(s_t, \pi(s_t))]$. We also define a stationary state occupancy distribution for $\pi$, denoted as $\rho^\pi(s) = (1 - \gamma) \sum_t \gamma^t p(s_t = s | \pi)$, and $\eta(\pi)$ can also be expressed as $\mathbb{E}_{s \sim \rho^\pi}[V_\pi(s)]$, where $V_\pi(s)$ is the state-value function.

**Attack model.** We follow existing attacks and assume the attacker injects *one* backdoor into the victim agent's policy. We add a superscript $'$ to indicate a backdoored/poisoned entity, and the backdoored policy is denoted as $\pi'$. We consider all three types of existing attacks: single-agent perturbation-based attacks [20], two-agent adversarial-agent attacks [49], and multi-agent perturbation-based attacks [8]. In the following, we use single-agent perturbation-based attacks to derive our technique, and later we discuss the extension of our approach to the other attacks. Under perturbation-based attacks, the attacker designs the trigger as a small perturbation patch denoted as $\hat{\Delta}$. At any time $t$, the attacker may add $\hat{\Delta}$ to the current state representation $s_t$, causing the backdoored agent to perceive the poisoned state $s'_t$ and taking a poisoned action $a'_t = \pi'(s'_t)$. We follow existing works [2, 20] and assume the attacker only perturbs the state representations perceived by the victim agent (victim agent's observation) without altering the actual state (underlying physical state). As such, the state transition still depends on the actual state rather than the perceived/poisoned one, i.e., $s_{t+1} \sim P(s_t, a'_t)$. To inject the backdoor, the attacker *needs to manipulate the victim agent's reward function, assigning the agent an ultra-high poisoned reward when it takes the poisoned action at poisoned states.*

**Defense assumption and goals.** We do not assume access to the attack training process. Instead, we are provided with an agent with a pretrained policy $\pi$ and the corresponding state-value network $V_\pi(s)$ or action-value network $Q_\pi(s, \pi(s))$, and lacks the knowledge of whether the policy is backdoored or not. We are also given a *clean environment* for testing and debugging purposes. This setup simulates a practical scenario where the defender needs to verify an agent's robustness and safety before deploying it in a potentially poisoned environment, especially in critical fields. We further assume access to limited computational resources that support testing or finetuning a given policy but not training a policy from scratch. We also do not assume any knowledge about attack specifications, including the ground-truth trigger, the poisoned actions, or the reward manipulation strategy employed by the attacker. We aim to develop a full-stack defense with the following three steps: 1) *Trigger restoration* - restore the potential trigger(s) from the given policy in the clean environment, 2) *Backdoor detection* - determining whether the restored trigger is associated with an injected backdoor and thus detect whether the given policy is backdoored or not, and 3) *Backdoor removal* - unlearning the detected backdoor from the backdoored policy without affecting its performance in the clean environment.

## 3.2 Trigger Restoration

**Key insights.** We derive the intuition from the training process of backdoor attacks. Recall that the attacker needs to poison the victim agent's reward function. Under the poisoned reward function, the agent receives a high total reward, denoted as $\eta(\pi') = \sum_s \rho^{\pi'}(s) \sum_{a'} \pi'(a'|s') R'(s', a')$, when taking trigger actions at the poisoned states. This indicates that given an agent with a fixed policy, adding a proper perturbation (trigger) to its state will activate the backdoor, resulting in a significant increase in the agent's total *poisoned* reward under the poisoned reward function.

Intuitively, we can restore the backdoor trigger by searching for a small perturbation to the states that *maximizes the agent's total poisoned rewards under the poisoned reward function.* Since we do not assume accessing to the poisoned reward function, we use the agent's state-value or action-value function to compute the total reward, which is also poisoned during the attack training. Formally, this perturbation can be obtained by solving the following objective function for a given policy $\pi$

$$\max_\Delta \sum_s \rho^\pi(s) \sum_a \pi(s + \Delta) Q_\pi(s + \Delta, \pi(s + \Delta)), \tag{1}$$

$\Delta$ has the same dimension as $s$, where each element in $\Delta$ represents the perturbation to the corresponding element in $s$. Solving Eqn. (1) yields a potential trigger for a backdoored policy but not for

a clean policy. Given that a clean agent's reward and value function is not poisoned, solving Eqn. (1) would likely produce a universal perturbation that guides the agent to take better actions to further increase its reward. However, this is typically challenging for complicated environments where the agent's task is difficult. Besides, we integrate specific designs to eliminate these noisy perturbations.

**Technical details.** We follow existing perturbation-based attacks [20, 8] and consider the state representation as the snapshot of the current environment, $s \in \mathbb{R}^{p \times q}$ with each element $s_{ij}$ is normalized to $[0, 1]$. Directly solving Eqn. (1) likely results in noisy perturbations for both backdoored and clean policies. To eliminate these noises, we propose two regularization terms to constrain the search space. First, given that the attacker typically uses a small trigger size to stay stealthy, we constrain $\mathcal{R}_1(\Delta) = ||\Delta||_1$, which yields a $\Delta$ with a small number of non-zero elements. This regularization rules out noisy perturbations with large sizes. Second, all existing perturbation-based attacks use dense patches as triggers because they are easily applicable in the physical world. In contrast, noisy perturbations are typically scattered. We add another regularization to constrain the smoothness/density of $\Delta$, $\mathcal{R}_2(\Delta) = \sum_{i,j} (\Delta_{i+1,j} - \Delta_{i,j})^2 + \sum_{i,j} (\Delta_{i,j+1} - \Delta_{i,j})^2$. This additional regularization term further rules out more noisy perturbations that are small and scattered.

Besides adding regularizations, we also design the perturbation $\Delta$ to be produced by a generative model. As demonstrated in previous works [5], assuming a generative model gives a more stable and less noisy result than directly solving the values in an unknown perturbation to neural network input. Since $s$ is normalized to $[0, 1]$, this constrains the maximum allowed perturbation to be 1 or $-1$, indicating $\Delta_{ij} \in [-1, 1]$. Since there is no common distribution with a value range of $[-1, 1]$, we satisfy this constraint by introducing a set variable $\{p_{ij}\}_{i=1:p, j=1:q}$. We define $p_{ij}$ to follow a Beta$(\alpha_{ij}, \beta_{ij})$ distribution, parameterized by $\alpha_{ij}$ and $\beta_{ij}$. Given that $p_{ij} \in [0, 1]$, we then compute $\Delta_{ij} = 2p_{ij} - 1$, which naturally guarantees $\Delta_{ij}$ lies within the range of $[-1, 1]$. The objective function can be rewritten as $\max_{\alpha_{1:p,1:q}, \beta_{1:p,1:q}} \mathbb{E}_{\mathbf{p} \sim \mathbf{B}} [\sum_s \rho^\pi(s) \sum_a \pi(s + 2\mathbf{p} - 1) Q_\pi(s + 2\mathbf{p} - 1, \pi(s + 2\mathbf{p} - 1))]$, where $\mathbf{p} \in \mathbb{R}^{p \times q}$ and $\mathbf{B}$ stands for the joint distribution $\prod_{i,j} \text{Beta}(\alpha_{ij}, \beta_{ij})$. By sampling perturbations from the beta distribution and computing the mean as the final result, we can reduce noise and obtain a trigger with higher fidelity. With this generative model, we double the number of parameters needed to be resolved. Given that the beta distribution's output sparsity depends more on the difference between $\alpha$ and $\beta$ than their actual value. We further fix the value of $\alpha$ and design $\beta_{ij} = \mathbf{e}_{ij} + \alpha, \mathbf{e} \in \mathbb{R}^{p \times q}$, to reduce the number of parameters in our objective function. Supplement S1 provides more insights about this parameter reduction design and Supplement S4 shows our method is not sensitive to the choice of $\alpha$.

So far, we consider the trigger's location and pattern to be fixed across different states (time steps). A straightforward attack adaption could change the trigger's location and even its shape at different states. To handle this scenario, we design $\mathbf{e}$ as a function of each state $f_\theta(s)$, parameterized by $\theta$. This design is also applicable to the original attack, where all the states have the same output for $f_\theta(\cdot)$. Putting all designs above together, we obtain the trigger restoration objective function.

$$\max_\theta J(\theta) = \mathbb{E}_{s \sim \rho^\pi} [\mathbb{E}_{\mathbf{P}_s \sim \mathbf{B}_s} [\eta_s(\pi(\mathbf{p}_s)) + \lambda_1 \mathcal{R}_1(\mathbf{p}_s) + \lambda_2 \mathcal{R}_2(\mathbf{p}_s)]], \quad \mathbf{B}_s = \prod_{i,j} \text{Beta}(\alpha, \alpha + (f_\theta(s))_{ij}),$$

$$\eta_s(\pi(\mathbf{p}_s)) = \sum_a \pi(s + 2\mathbf{p}_s - 1) Q_\pi(s + 2\mathbf{p}_s - 1, \pi(s + 2\mathbf{p}_s - 1)),$$

(2)

where $\lambda_1$ and $\lambda_2$ are hyperparameters. $\mathbf{p}_s$ and $\mathbf{B}_s$ is specific for each state $s$. Eqn. (2) can not be directly solved because $\mathbf{p}_s$ is sampled from an unknown distribution $\mathbf{B}_s$. We leverage the policy gradient method [21, 18] to compute the gradient $\nabla_\theta J(\theta)$ and solve the Eqn. (2) following the typical on-policy REINFORCE algorithm [41]. Our final perturbation for each state $s$ is $\Delta_s = 2(\frac{\alpha}{2\alpha + f_\theta(s)}) - 1$, where $\frac{\alpha}{2\alpha + f_\theta(s)}$ is the mean of the joint beta distribution $\mathbf{B}_s$.

To ensure that the perturbations we compute actually influence the agent's actions and affect its reward during execution, rather than generating simple adversarial examples for the value network that cannot affect the agent's actions, we only use the agent's value network to compute average rewards. We do not compute the gradient of $\theta$ with respect to the value network or update the value network itself. Supplement S1 provides a detailed algorithm for solving Eqn. (2) and also discusses further why we are not solving adversarial examples for the value network.

### 3.3 Backdoor Detection

**Key insights.** Our backdoor detection methods draw insights from the unique behaviors of backdoored policies, which perform well in clean environments but poorly in poisoned ones. For a backdoored

policy, if its restored trigger $\Delta$ has similar efficacy as the ground-truth trigger $\hat{\Delta}$, adding it back to the clean environment will significantly reduce the agent's total actual reward. However, this is not the case for a clean agent trained using the actual reward function. Since the value function of a clean agent is not poisoned, solving Eqn. (2) will either result in a perturbation that guides the agent to select better actions and increase its reward or a perturbation that cannot significantly impact the agent's performance, if the agent is already well-trained or the environment is too complicated. In both cases, adding the perturbation to the environment will not reduce the agent's actual reward.

As such, we propose a novel metric to distinguish between backdoored and clean policies, which is *based on the agent's actual reward reduction before and after poisoning the environment with restored perturbations*. This approach differs from existing detection methods designed for supervised classifiers (e.g., [4, 48, 38, 42]), which rely on the trigger size ($\|\Delta\|_0$) as the metric. This metric is sensitive to the ground-truth trigger size. Our detection method is designed based on the unique behaviors of backdoor attacks in DRL and is more accurate and robust (demonstrated in Section 4).

**Technical details.** Given an agent's policy $\pi$, we first run it in the environment for $K$ rounds and record the agent's average actual reward given by the environment, denoted as $\bar{\eta}(\pi) = \frac{1}{K} \sum_t^{(k)} R(s_t^{(k)}, \pi(s_t^{(k)}))$. We then simulate an extreme attack case by adding the restored perturbation $\Delta_s$ to the agent's policy input at each time step, run the agent for $K$ rounds, and record the average actual reward, denoted as $\bar{\eta}(\pi, \Delta) = \frac{1}{K} \sum_t^{(k)} R(s_t^{(k)}, \pi(s_t^{(k)} + \Delta_{s_t}))$. For adversarial agent attacks, since $\Delta$ corresponds to the state representation of the adversarial agent's trigger action, adding $\Delta_{s_t}$ to $s_t$ simulates the trigger action. Here, $\bar{\eta}(\pi, \Delta) = \frac{1}{K} \sum_t^{(k)} R(s_t^{(k)} + \Delta_{s_t}, \pi(s_t^{(k)} + \Delta_{s_t}))$, given that the trigger action indeed changes the actual state of the environment. Next, we compute the reward reduction rate $\phi(\pi, \Delta) = (\bar{\eta}(\pi, \Delta) - \bar{\eta}(\pi))/\eta_{\max}$, where $\eta_{\max}$ is a normalization term, representing the maximum reward difference of the environment. $\phi(\pi, \Delta) \in [-1, 1]$. If $\Delta$ is similar to the ground-truth trigger, it will result in a very large reward reduction and thus a negative $\phi(\pi, \Delta)$ near $-1$. Conversely, a $\Delta$ restored from a clean policy will produce a positive $\phi(\pi, \Delta)$ or a $\phi(\pi, \Delta) \approx 0$.

Therefore, we treat $\pi$ as a backdoored policy and $\Delta$ as the trigger if $\phi(\pi, \Delta) \leq \epsilon$. We consider a practical scenario where we do not have a set of clean policies to help select a proper threshold. Instead, we need to make a decision for each individual policy by hand. We set an aggressive value with $\epsilon = -0.9$ to rule out false positives. Here, we consider all of the clean policies and backdoored policies are well-trained, i.e., their reward in the clean environments are nearly optimal. In case that there are some sub-optimal agents that are backdoored, the detection threshold should be more conservative. More discussions on the detection threshold under different situations can be found in Supplement S2. Later, in Section 4, we demonstrate that, due to the significant performance difference between backdoored policies and clean policies under our restored perturbation, even setting $\epsilon = -0.9$ introduces only negligible false negatives and thus provides a nearly perfect detection performance. Supplement S1 presents our detection algorithm.

### 3.4 Backdoor Removal through Unlearning

**Naive solutions and pitfalls.** A naive solution for backdoor removal is to add the restored trigger to the state and retrain the policy under the actual reward function, which penalizes poisoned actions $a'$. We test this solution and find that it indeed improves the agent's performance when presented with the restored trigger. However, the retrained policy remains vulnerable to the ground-truth trigger due to differences between the restored and ground-truth trigger. We try to address this issue by varying the restored

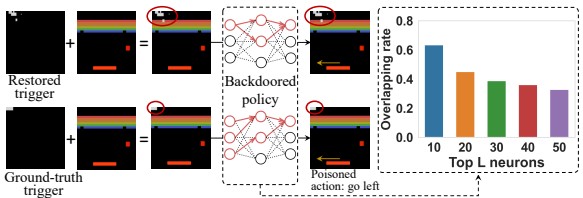

Figure 1: Overlapped shortcuts between the restored trigger and the ground truth one. The bar plot shows the overlapping percentage among the top $X$ neurons with the highest value.

trigger (e.g., adding random noise) during the training. Our empirical findings suggest that the ground truth trigger is only unlearned when the restored one is very similar to it, which is not always the case. Furthermore, we discover that the performance of retrained policies in the clean environment oftentimes decreases after retraining. These initial explorations highlight two challenges: (1) how to remove the ground-truth trigger when the restored one has some differences from it, and (2) how to maintain the retrained agent's performance in the clean environment.

**Our Designs.** As extensively studied in existing research [54, 53, 10], injecting a backdoor in neural networks is equivalent to building one or a few *shortcuts* from the backdoor trigger to the target out. In DRL, these shortcuts connect the poisoned states with the poisoned action. When the trigger is presented, only the neurons on the shortcuts get activated. Given that our restored trigger could indeed trigger similar poisoned actions as the ground truth one, we hypothesize that even if the restored trigger is not highly similar to the ground truth one, their corresponding shortcuts may have some overlap. To test this hypothesis, we conduct an empirical test where we compared the neurons activated by the restored trigger with those activated by the ground truth trigger. As shown in Fig. 1, the neurons with high activation values have a large overlap for these two triggers, confirming our hypothesis. Guided by this intuition, we propose to remove these most vulnerable and overlapping shortcuts instead of directly unlearning the trigger. In particular, we first add the restored trigger to the backdoored policy input and record the activation values of all the neurons. We select the top $L$ neurons with the highest activation value and re-initialize their weights to erase the most vulnerable shortcuts. Finally, we retrain the policy in the environment poisoned with the restored trigger under the actual reward function. The objective function can be written as $\max_\phi \eta(\pi_\phi, \Delta)$, where $\pi_\phi$ denotes the retrained policy. We retrain the entire policy network rather than only the reset neurons to give the policy enough capacity to let the agent learn how to perform in the poisoned environment. See Supplement S2 for more details about our neuron selection and re-initalization process.

The second challenge can be addressed by incorporating an additional regularization term, i.e, $|\eta(\pi_\phi) - \eta(\pi')| \leq \epsilon_1$, where $\pi'$ is the original backdoored policy. This ensures that the total reward difference between the retrained policy and the original policy is within a certain threshold $\epsilon_1$. This constraint helps to maintain the retrained agent's performance in the clean environment, given that the original backdoored agent should have performed well in a clean environment by design. However, this regularization is not feasible as $|\eta(\pi_\phi)|$ is intractable during training. To overcome this, we propose to use the KL divergence between $\pi_\phi$ and $\pi'$ as a proxy since it is much easier to compute. The following inequality states the relationship between KL divergence and value difference.

$$|\eta(\pi_\phi) - \eta(\pi')| \leq C \max_{s \sim \rho^{\pi'}} \mathbb{KL}(\pi_\phi(s) \| \pi'(s)). \qquad (3)$$

See Supplement S1 for the derivation. Based on Eqn. (3), constraining the KL divergence between $\pi$ and $\hat{\pi}$ can bound $|\eta(\pi_\phi) - \eta(\pi')|$. Then, our final retraining objective function can be written as

$$\max_\phi \eta(\pi_\phi, \Delta) \,, \text{s.t. } \mathbb{KL}(\pi_\phi(s) \| \pi'(s)) \leq \epsilon_1 \,. \qquad (4)$$

We leverage the state-of-art PPO algorithm [37] to solve Eqn. (4), which guarantees a fast convergence. During retraining, we periodically add $\Delta$ to simulate the poisoned states and keep a certain portion of clean states to compute $\mathbb{KL}(\pi_\phi(s) \| \pi'(s))$. Supplement S1 shows the detailed algorithm of our backdoor removal methods. Supplement S2 states our implementation and hyper-parameter choices.

Note that our proposed defense (from trigger restoration to backdoor removal) can be directly applied to adversarial agent attacks [49] and multi-agent perturbation-based attacks [8] (See Supplement S1 for a detailed discussion).

## 4 Evaluation

We compare BIRD with three state-of-the-art backdoor DRL attacks: TrojDRL [20], Backdoorl [49], and Marnet [8]. Following their papers, we select the environments and used their default attack setups to train backdoor policies (see Supplement S2 for more details). For TrojDRL, we select six Atari games from the OpenAI Gym [3]: Breakout, SpaceInvaders (SI), Qbert, and Seaquest, CrazyClimber, Pong. Due to limited space, we put the results of CrazyClimber and Pong in Supplement S3. TrojDRL considers targeted and non-targeted attacks based on whether the backdoored action is prespecified or not. In this section, we report the results of non-targeted attacks, where detection is more challenging, and leave the results of targeted attacks in Supplement S3. For Backdoorl, we choose the You-Shall-Not-Pass (YSNP), Sumo-Humans (SH), and Run-To-Goal-Ants (RTGA) games from the MuJoCo environment [45]. We select the SMAC environment [47] for Marnet and conduct two attacks, one against the policies trained by QMIX [35] (a multi-agent Q-learning algorithm), and the other against the policies trained by COMA [12] (a multi-agent policy gradient algorithm). Note that by default, all the perturbation-based attacks consider the trigger with a fixed shape/size/location, and adversarial agent attacks set the trigger as a sequence of fixed actions taken at the beginning of each game. We include the performance of BIRD against a more recent attack [6] in Supplement S3.

Table 1: Performance of different defenses in the nine attack scenarios. "Clean" and "poisoned" refer to the original clean environment and an environment poisoned by the ground truth trigger. The numbers in each element are mean±std. "-" means that the method is not applicable to the corresponding attack. The Atari games' results are rewards and the rest are winning rates. Supplement S3 shows the p-value for each attack scenario is smaller than 0.001.

| Environments | Methods | Breakout | SpaceInvaders | Qbert | Seaquest | QMIX | COMA | YSNP | SH | RTGA |
|---|---|---|---|---|---|---|---|---|---|---|
| Clean | Original | 298±37 | 610±97 | 13488±1897 | 1719±107 | 74.4%±2.5% | 93.0%±2.2% | 52.8%±4.5% | 28.9%±3.6% | 49.7%±4.8% |
| | PD | 142±32 | 238±94 | 4941±1428 | 932±97 | - | - | - | - | - |
| | Finetuning | 317±39 | 614±104 | 13151±2233 | 1785±98 | 73.6%±2.6% | 90.5%±5.4% | 53.5%±1.7% | 30.1%±2.8% | 50.6%±4.4% |
| | BIRD | 320±35 | 529±92 | 13389±2452 | 1741±73 | 70.2%±1.3% | 89.2%±3.8% | 48.2%±1.8% | 27.2%± 1.3% | 44.2%± 3.2% |
| Poisoned | Original | 8.0±4.8 | 19±87 | 150±12 | 146±49 | 6.5%±1.4% | 1.3%±2.7% | 16.5%±3.4% | 11.2%±2.8% | 22.5%±1.8% |
| | PD | 140±9 | 325±53 | 6180±1817 | 853±162 | - | - | - | - | - |
| | Finetuning | 10.6±0.9 | 21±65 | 173±24 | 159±63 | 6.7%±1.5% | 5.31%±8.3% | 18.4%±2.1% | 12.5%±3.3% | 21.5%±1.6% |
| | BIRD | 269±49 | 543±52 | 12694±2194 | 1686±83 | 70.9%±4.8% | 90.7%±3.0% | 46.7%±3.9% | 25.1%±2.6% | 43.1%±6.5% |

## 4.1 Trigger Restoration and Backdoor Detection

**Baselines.** There are no existing works on detecting backdoored policies.[1] Therefore, we select two backdoor detection techniques designed for supervised classifiers. The first is the widely used NC method [48], and the other is the state-of-the-art method, Pixel [42]. They can be applied to perturbation-based attacks (TrojDRL and Marnet), where the action space is discrete. In particular, we collect a set of states and actions of a given policy and run these methods by treating each possible action as the target class. We then apply their detection method (i.e., using the trigger's $l_0$ norm as the metric to find if there is an outlier trigger) to decide whether the policy is backdoored or not.

**Design and Metric.** For each of the nine attack scenarios, we trained 10 policies, with five of them being backdoored policies, five being well-trained clean policies. We applied BIRD (with $\epsilon = -0.9$) and the baseline methods to identify the backdoored policies from the 10 policies, and reported the F1 score of the detection result. Both BIRD and the baseline methods make the detection decision for each individual policy without relying on other policies. Therefore, the total number of policies does not affect the detection process.

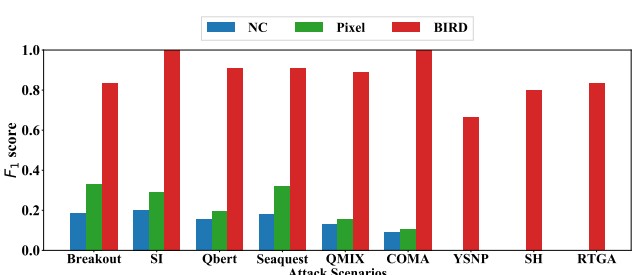

Figure 2: Backdoor detection F1 scores of the selected methods in the nine attack setups. "SI" stands for the Atari SpaceInvaders environment. "QMIX" and "COMA" stands for the attack against QMIX and COMA policies in the SMAC environment. Given that NC and Pixel cannot be applied to adversarial agent attacks, we do not report their results for those attacks.

**Results.** Fig 2 shows the backdoor detection performance of all three methods in the nine attack scenarios. The figure first shows that the baseline methods perform poorly in all the attacks where they can be applied. This result confirms that without considering the sequential decision-making nature of DRL, these baseline methods have limited generalizability in detecting DRL backdoor attacks. In contrast, BIRD has a much higher (in some cases perfect or near perfect) F1 score in each setup, demonstrating the effectiveness of our method and its superior advantage over existing methods. As we will elaborate in Section 4.3, both our trigger restoration technique and novel detection metric contribute to this remarkable result. Additionally, we show in Supplement S3 that the triggers obtained from BIRD have higher fidelity than those restored by the comparison baselines.

## 4.2 Backdoor Removal

**Baselines.** We select two baseline methods: 1) direct finetuning of the backdoored policy in the corresponding clean environment, denoted as "Finetuning", and 2) an existing DRL backdoor defense

---

[1]Existing DRL defense method [2] applies their defense procedure to each policy without distinguishing whether it is backdoored or not.

Table 2: Ablation study results. "NR" refers to neuron re-initialization. For $||\Delta||_0$, we carefully tune the threshold and report the best result.

| Attack scenario | F1 score | | Poisoned | |
|---|---|---|---|---|
| | $\phi(\pi, \Delta)$ | $||\Delta||_0$ | w. NR | w/o. NR |
| Qbert | 0.909 | 0.571 | 12694±2194 | 0.0%±0.0% |
| COMA | 1.000 | 0.347 | 90.7%±3.0% | 6.7%±2.1% |
| YSNP | 0.667 | 0.263 | 46.7%±3.9% | 29.5%±4.1% |

Table 3: BIRD against attack variations. Each column represents one variation (e.g., Col.3 means circle, $3 \times 3$, 0.1). "Clean" and "Poisoned" refer to the performance in clean and poisoned environments.

| | Default setup | Shape | | Size | | PR | |
|---|---|---|---|---|---|---|---|
| | | circle | triangle | $5 \times 5$ | $7 \times 7$ | 0.2 | 0.4 |
| F1 score | 1.0 | 0.909 | 1.0 | 0.833 | 0.889 | 1.0 | 0.889 |
| Clean | 529±29 | 536±68 | 609±36 | 601±85 | 636±93 | 540±56 | 471±83 |
| Poisoned | 543±52 | 501±47 | 575±24 | 557±61 | 629±72 | 544±61 | 428±33 |

designed for single-agent perturbation-based attacks - Provable Defense (PD) [2]. We do not consider another existing defense [15] as it is still a pre-print paper without a public implementation. We also do not include the large body of backdoor unlearning techniques designed for supervised classifiers [34, 25, 24, 61, 39, 68], which cannot be directly applied to our problem.

**Design and Metric.** For each attack scenario, we apply the selected methods to defend the 5 backdoored policies trained in Section 4.1. We use the agent's average reward/winning rate over 1000 game rounds as the evaluation metric. We report the mean and standard deviation of the metrics over the five policies in the clean and poisoned environment before and after applying the defense. We construct the poisoned environment by adding the ground truth trigger to every state. To demonstrate the statistical significance of our results, we also conduct a paired t-test to compare BIRD with the selected baselines and report the p-value. For BIRD , we select the number of reinitialized neurons $L \in [10, 30]$ for each policy based on the retraining performance in the first few iterations. We reinitialize the values of the selected neurons as zero.

**Results.** Table 1 presents the defense results of the selected methods. First, we observe that policies directly fine-tuned in clean environments still perform poorly in the poisoned environment, indicating that the backdoor is not removed. Second, PD demonstrates certain defense efficacy, as it increases the agent's reward in the poisoned environments. However, PD significantly reduces the agent's performance in clean environments. We suspect that this is due to the fact that PD project s every state to a possible clean state. This is problematic when PD cannot correctly project the organically clean states, which lowers the quality of the projected states and thus reduces the agent's performance in clean environments. Additionally, PD involves computing singular value decomposition when conducting the projection, which is computationally expensive. The average run time of PD is at least $2 \times$ larger than BIRD. Finally, the policies retrained by BIRD demonstrate the highest reward/winning rate in poisoned reward across all attacks, which is almost similar to the original backdoored agent's performance in the clean environment, verifying BIRD successfully removed the backdoor. Moreover, BIRD can well maintain the agent's performance in a clean environment. In conclusion, BIRD is much more effective, efficient, and generalizable than existing defenses in robustifying a backdoored policy and maintaining its utility in a clean environment. In addition, we also show that BIRD outperforms two other alternative defenses, where we use the trigger restored by NC and Pixel and apply our backdoor removal method to retrain the backdoored policies (See Supplement S3).

## 4.3 Ablation Studies

In this experiment, we conduct three ablation studies to verify the efficacy of our three key designs: generative model (Section 3.2), detection metric (Section 3.3), and neuron re-initialization (Section 3.4). We select one game from each attack for this experiment: Qbert (non-targeted attack) and COMA for the perturbation-based multi-agent attack and YSNP for the adversarial agent attack. In the following, we discuss our ablation studies on the detection metric and neuron re-initialization while leaving the experiment on the generative model to Supplement S3.

**Detection metric.** For each attack scenario, we reuse the ten policies and their restored triggers from Section 4.1. We then replace our reward-based metric $\phi(\pi, \Delta)$ with the commonly used metric – $||\Delta||_0$ and deem the policy with $||\Delta||_0$ smaller than a certain threshold as the backdoored one. Table 2 (Col. 2&3) shows that using our proposed metric leads to significantly higher detection rates than $||\Delta||_0$, verifying the effectiveness of our design. As mentioned in [48, 16], detecting backdoors using trigger size is difficult because noisy perturbations or adversarial examples can also have a small size. In contrast, our metric is designed based on behavioral differences between clean and backdoored policies, making it more robust to noisy perturbations. Furthermore, BIRD are insensitive to the threshold $\epsilon$, significantly improving its practicality (See Supplement S4).

**Neuron Re-initialization.** We reuse the five backdoored policies from the three attacks and compare the effectiveness of our proposed neuron re-initialization method against directly retraining the entire policy network with the restored trigger using Eqn. (4). Table 2 (Col. 4&5) shows the retrained policies' performance in the environment poisoned by the ground truth trigger with and without neuron re-initialization. Comparing Col. 2&3 can reflect the efficiency of using total reward as the detection metric and comparing Col. 4&5 can reflect the efficiency of neuron re-initialization. Our results demonstrate the importance of neuron re-initialization in backdoor removal. This also confirms our claim in Section 4.2 that directly retraining the policy with the restored trigger may not fully eliminate the ground truth trigger when they are not very similar.

### 4.4 Attack Variations and Adaptive Attacks

We evaluate the robustness of BIRD against different attack variations and two potential adaptive attacks in the SpaceInvaders environment of the non-targeted perturbation-based attack.

**Attack variations.** Based on the default attack setup, where the trigger is a $3 \times 3$ square, and the state poisoning rate is $0.1$, we vary the trigger shape (circle and triangle), size ($5 \times 5$, $7 \times 7$), and poisoning rate ($0.2, 0.4$). For each variation, we train five backdoored policies, apply BIRD to detect them from the five clean policies mentioned in Section 4.1, and report the detection performance in Table 3. The results show that BIRD maintains a high and stable detection performance when dealing with these variations. Table 3 also presents the backdoor removal performance of BIRD on these attacks, which is consistent with the detection result.

**Adaptive attack-1.** We first consider a straightforward adaptive by allowing the trigger to vary its shape and locations at different states. However, even after carefully tuning the training parameters, the attack was unable to achieve comparable performance as it is in the static setting. For SpaceInvaders, the agent's average reward in the poisoned environment is only reduced from 580 to 510 for non-targeted attack and (to 445 for targeted attack). This motivates future work to design more effective backdoor attacks with dynamic triggers. Nevertheless, even if such an attack succeeds, as discussed in Section 3, BIRD is designed to be robust against it.

**Adaptive attack-2.** We further consider an adaptive attack against our neuron re-initialization mechanism, a key design for backdoor removal. When training the backdoored policy, we add an additional regularization to constrain the $l_2$ norm of the policy weights, which avoids neurons with ultra-high activation values. We train five backdoored policies with this attack, apply BIRD , and record the backdoor removal performance. The policies' average reward in the clean environment before and after applying BIRD is 582 and 563, respectively, and the average reward in the poisoned environment is 170 (before) and 512 (after). These results demonstrate the robustness of BIRD against this adaptive attack (See Supplement S2 for more implementation details).

Additionally, we discuss generalizing BIRD to a setup where we do not assume accessing the agent's value networks, or a more challenging scenario where the attacker intentionally attaches a benign value network instead of the backdoored one to make the attack more stealthy(See Supplement S3). Moreover, we also demonstrate the computational efficiency of BIRD and its insensitivity to subtle variations in key hyper-parameters ($\alpha$, detection threshold $\epsilon$, and the number of reinitialized neurons $L$). We present these experiments in Supplement S4.

## 5 Conclusion and Future Works

We present BIRD, a method for detecting and removing backdoors in DRL policies. By analyzing the attack process and unique behaviors of backdoor agents, we formulate the trigger restoration as an optimization problem and design a novel metric to detect backdoored policies. We also propose a novel finetuning method to remove the identified backdoor. Extensive experiments against various attacks demonstrate the effectiveness, efficiency, and generalizability of BIRD, as well as its robustness to different attack variations.

This work points to several promising directions for future research. First, we evaluate two adaptive attacks in Section 4. Our future work will explore other possible attacks (e.g., using watermark triggers) and extend BIRD to defend against these attacks. Second, while our trigger restoration technique may not always yield a restored trigger that perfectly matches the ground-truth trigger, it is still effective in achieving a decent detection and removal performance. We plan to refine our

trigger restoration technique to improve its restoration fidelity in future work. Third, in addition to the detection and unlearning mechanism utilized in BIRD, we will explore alternative solutions to render the backdoor ineffective. For example, we can filter out the trigger before feeding the state representation to the backdoored agent without finetuning it. Fourth, we acknowledge that besides the neuron re-initialization method discussed in Section 3.4, existing works also design other techniques to identify and remove the backdoor shortcuts (e.g., neuron pruning [27, 53, 58], [51]). In future work, we will explore the effectiveness of integrating these techniques into our approach. Fifth, as discussed in Section 2, BIRD differs from other robust RL methods in terms of attack/problem setups. As part of future work, we will investigate generalizing these defenses designed for other attack setups to defend against backdoor attacks. Finally, while we evaluate BIRD in various game environments, we plan to further extend the backdoor attacks and defenses to broader types of environments, such as multi-agent competitive environments [29, 43] and extensive-form environments [23, 1].

## Acknowledgements

This project was supported, in part by ARL Grant W911NF-23-2-0137, IARPA TrojAI W911NF-19-S0012, NSF 1901242 and 1910300, ONR N000141712045, N000141410468 and N000141712947. We also thank the Center for AI Safety for their support of the Compute Cluster. Any opinions, findings, and conclusions in this paper are those of the authors only and do not necessarily reflect the views of our sponsors.

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
