# OpenReview forum: "BIRD: Generalizable Backdoor Detection and Removal for Deep Reinforcement Learning"
_NeurIPS.cc/2023/Conference — NeurIPS 2023 poster_

### Official Review · Reviewer_M2QE · 2023-07-06

**Soundness:** 2 fair
**Presentation:** 2 fair
**Contribution:** 2 fair
**Rating:** 5
**Confidence:** 5

**Summary:**

The paper addresses the challenge of detecting backdoored reinforcement learning policies. The injection of backdoors in RL policies was first studied in [19] and while there has been some work on detection of backdoored policies in [2] and [14] - these methods are limited to settings where the trigger is in the competing agent's actions [14] or limited to perturbation patches [2]. This paper formulates trigger restoration as an optimization problem and designs a novel metric to detect backdoored policies.

**Strengths:**

The paper attempts to solve an important problem on supply chain of machine learning models - detection of backdoored policies and elimination of triggers.

**Weaknesses:**

* The formulation has one serious weakness - it assumes that we have access to value function in addition to the agent's policy network and the clean environment. The value function is "poisoned" and so, just optimizing against this value function allows one to discover triggers. This is unrealistic. In practice, the RL policies in the supply chain would be procured as a state to action map or sequence of state to a action - that is, one would just have access to the policy and not have access to the value function.

The entire approach is predicated on this availability of the value function because the trigger recover is framed as an optimization problem that maximizes the value function. So, this is the primary concern of the reviewer.

* Some statements are inaccurate though they are not serious concerns and are more of presentation issue. For example, line 108/109, "TO inject the backdoor, the attacker needs to manipulate .. reward function .. " ... This is just one way to inject Trojans - for supervised learning, one can inject Trojans by directly manipulating the network and the same could be feasible for RL. So, it would be better to say, "existing methods manipulate reward function for injecting backdoor.."

* The method for trojan elimination uses a number of hyperparams which appear to be critical for success of the method - definition of high activation for a neuron, selection of L top neurons, threshold epsilon_1, etc. In its current presentation, the approach looks rather hacky that might be made to work on 9 scenarios but will be difficult to generalize.

**Questions:**

* Can you help understand a practical setting where one would necessarily have access to policy as well as the value function for the policy ? Isn't this a very major assumption and severely limits the use of the proposed method? It also makes the problem much more easier to solve. Even if a buyer forces that policy be accompanied by value function -the attacker could produce value function with the trigger removed. Why would an attacker provide a value function that has the trigger behavior embedded in it?

* The part of different triggers having common shortcut is very interesting. But Fig 1 is not clear. What does "high activation" mean here? Neurons can have very different scales of activation, particularly across different layers - how do you decide that some activation is high and compare different neurons with each other? What does it mean to reinitialize weights for the highest value neurons - how does it correspond to breaking shortcuts (traditional definition of shortcut is that low layer neurons impacting very high layer neurons).



**Limitations:**

There are no concerns of negative broader impact.

---

> ### Author Rebuttal · Authors · 2023-08-07
>
> We thank the reviewer for their constructive review. Please see below for our response and clarifications.
>
> **The reviewer first questioned the assumption of accessing the target agent’s value function.**
>
> We thank the reviewer for the valuable feedback. We would like to kindly point out that this assumption can be practical in many cases. Our assumption simulates a practical scenario that the agent user outsources the policy training process to a third party (model training service provider). The user can request to get both the policy network and value function from the service provider or only choose the provider that is willing to provide the value function. As for service providers, it does not introduce the extra cost of providing the value function to their customers as they will need to train it anyway.  For example, we checked the top 10 models that are most popular in **Hugging Face**, and we found that all of them included the value network.
>
> We also agree with the reviewer that the attacker could intentionally attach a benign value network instead of the backdoored one to make the attack more stealthy, or we do not have access to the value network. We address this scenario and provide a corresponding solution in Supplement S3.4. In such cases, we restore the trigger by minimizing the actual return of the agent (i.e., change the objective function in Eqn. (1) to
>
> $\min_\Delta  \sum_s\rho^{\pi}(s)\sum_{a}\pi(s+\Delta)R(s+\Delta, \pi(s+\Delta)))$,
>
> where $R$ is the actual reward function of the RL problem. The insight is for a backdoored agent, its actual return will drop when facing the triggered environment.
>
> Fig. S4 in Supplement shows that our method can still outperform all baselines in backdoor detection. Our removal step is not affected as Eqn. (4) does not require the agent’s value function. It optimizes the agent’s total return under the actual reward. Table 4 in the submitted pdf file shows that with the trigger restored using this adaptive method, our backdoor removal is still effective (We select five setups from the nine attack scenarios). We will highlight this in the next version.
>
> **The reviewer then questioned the sensitivity of our method to key hyper-parameters.**
>
> Thank the reviewer for the valuable feedback. In Supplement S4, we systematically evaluate the sensitivity of BIRD to hyper-parameter changes, including the one pointed out by the reviewer – the number of neurons/convolutional kernels selected in each layer for resetting ($L$) in Table S8. The results indicate that our approach is insensitive to the variations in these hyper-parameters.
>
> In addition, we conduct an additional experiment to evaluate the $\epsilon_1$ in Eqn.(4). Table 5 in the submitted pdf file shows that our method is robust against the variation of this hyper-parameter. We will add this result in the next version.
>
> **The reviewer then asked for clarifications about our neuron reinitialization process and its connection to breaking shortcuts.**
>
> Thank you for pointing this out. Sorry for the confusion.
>
> We notice that the activation values of neurons at different layers are at different scales. As such, we sort neurons based on their activation value within each layer. Our policy networks involve two types of layers – the convolutional layer and the fully-connected layer. More specifically, as shown in Algorithm 3 in Supplement S1, during the backdoor removal, we first run a warm-up stage to collect a set of $E$ trajectories from the poisoned environment. For each state in these trajectories, we input it into the policy network and record the activation value for each kernel in the convolutional layer and each neuron in the feedforward layer. We then compute the activation mean of each kernel/neuron across all the states.
>
> Given these mean activation values, we then select $L$ neurons/kernels from each layer in the policy network. Specifically, for each convolution layer, the output for each state is a 3-D tensor with the size of $[C, H, W]$. We first find the highest activation value in each channel (i.e., each $H\times W$ 2-D activation matrix). Then, we rank these channel-wise highest activation values to select the top $L$ channels with the $L$ highest values. We reinitialize the weights and biases of the kernels corresponding to the selected channels. For each linear layer, we rank the activation value and select the top $L$ neurons. We reinitialize the weight and bias of these selected neurons. Regarding the reinitialization operation, we reset the weights and biases as zero. We are again sorry for the confusion due to these missing details. We will add them together with a more detailed description of Algorithm 3 in the next version.
>
> We came up with this idea following [1][2]. As discussed in [1], when the trigger is presented in the input, the neurons/kernels with high activation values at each layer typically form the backdoor shortcuts. Resetting their weights and biases as zero can potentially remove the shortcuts and give the model flexibility in learning the correct actions under the poisoned inputs. In our method, we follow this idea to remove backdoor shortcuts and give the policy the opportunity to learn correct policies in the poisoned environment.
>
>
> [1] Training with More Confidence: Mitigating Injected and Natural Backdoors During Training, NeurIPS 2022.
>
> [2] Adversarial Neuron Pruning Purifies Backdoored Deep Models, NeurIPS 2021.

---

> > ### Comment · Reviewer_M2QE · 2023-08-13
> > **Thank you for the supplementary results**
> >
> > "We also agree with the reviewer that the attacker could intentionally attach a benign value network instead of the backdoored one to make the attack more stealthy, or we do not have access to the value network. We address this scenario and provide a corresponding solution in Supplement S3.4. "
> >
> > Yes, this answers my concern. The authors might want to expand the scope of the presentation, and not make it appear too reliant on value network access.
> >
> > "Thank the reviewer for the valuable feedback. In Supplement S4, we systematically evaluate the sensitivity of BIRD to hyper-parameter changes, including the one pointed out by the reviewer – the number of neurons/convolutional kernels selected in each layer for resetting (
> > ) in Table S8. The results indicate that our approach is insensitive to the variations in these hyper-parameters."
> >
> > Thank you. This addresses my second major concern.
> >
> > I will raise my score to be positive.

---

> > > ### Author Response · Authors · 2023-08-13
> > >
> > > We do appreciate the reviewer's thoughtful review of our paper and the positive feedback.
> > >
> > > We are pleased to hear that our solution in Supplement S3.4 addresses the reviewer's concern regarding the assumption of the value network. And we will certainly work on enhancing this aspect to provide a more comprehensive view of our approach.
> > >
> > > Moreover, we are grateful that our evaluation of hyper-parameter sensitivity addresses the reviewer's second major concern.
> > >
> > > We would like to thank the reviewer for increasing the score. As we proceed with the revision, we will be mindful of the suggestions and present a stronger version of our paper based on the reviewer's feedback.

---

### Official Review · Reviewer_jZ8t · 2023-07-07

**Soundness:** 3 good
**Presentation:** 3 good
**Contribution:** 2 fair
**Rating:** 5
**Confidence:** 3

**Summary:**

This paper addresses the threat of backdoor attacks against deep reinforcement learning (DRL) policies. To tackle this problem, the authors propose BIRD, a novel generalizable backdoor detection and removal method for pretrained DRL policies in a clean environment without any knowledge of the attack specifications or access to the training process. They formulate trigger restoration as an optimization problem and introduce a novel metric for detecting backdoored policies. The authors also develop a fine-tuning method to remove the detected backdoor. The environmental results demonstrate the effectiveness and computational efficiency of BIRD, as well as its robustness against various backdoor attacks.

**Strengths:**

1. Originality: BIRD, is the first approach to detect and remove backdoors from a pre-trained DRL policy without requiring any prior knowledge. The authors demonstrate the novelty and advantages of each component of the algorithm, highlighting the strong practical value of the proposed technique in addressing backdoor attacks.
2. Quality and clarity: The paper is well-written and organized, effectively conveying the authors' motivation and the details of the proposed technique. The main contribution lies in the development of a comprehensive defense strategy consisting of trigger restoration, backdoor detection, and removal. The authors provide clear and detailed insights into each part, along with the technical details. The experiments are well-designed, conducted rigorously, and include ablation studies to validate the efficacy of key design choices in BIRD.
3. Significance: By providing an effective defense approach, the authors contribute valuable insights to adversarial RL defense against backdoor attacks. This work opens up possibilities for further research and development in addressing backdoor attacks, making it a notable contribution to the community.

**Weaknesses:**

1. Based on my understanding of this paper, the authors' approach focuses on the detection and removal of backdoor attacks rather than training a robust policy from scratch to defend against them. BIRD performs fine-tuning to remove the backdoor, but it does not directly improve the robustness of the RL algorithm itself. However, I do not have any issues with the authors' approach; I am simply highlighting this aspect.

2. Regarding the presentation of experimental results, I noticed that the tables can sometimes be confusing. For instance, in Table 2, directly comparing the results of Qbert, COMA, and YSNP may lead to misunderstandings.

3. I am curious to know whether the choice of different backdoor attack methods would also impact the effectiveness of BIRD. Could the authors provide a brief explanation of this question?

**Questions:**

In summary, I have few additional questions or concerns. As mentioned in the weaknesses section, there were some suggestions for improvement provided. I am willing to revise my assessment after further discussion.

**Limitations:**

The paper extensively discusses the future directions of their approach, exploring the potential applications, improvements, and extensions of BIRD. However, it would be beneficial to include more discussions regarding the limitations and practical significance of the proposed method. The potential impact of the work should be better highlighted, offering insights into its practical implications.

---

> ### Author Rebuttal · Authors · 2023-08-07
>
> We thank the reviewer for their positive and constructive review. Please see below for our response and clarifications.
>
> **The reviewer first questioned the presentation of Table 2.**
>
> We thank the reviewer for the suggestion.  We agree with the reviewer that the results of Qbert, COMA, and YSNP can not be compared directly. In Table 2, we show the ablation study results of two key designs in our method: our proposed reward based detection metric and the neuron re-initialization strategy. Comparing Columns 2&3 and Columns 4&5 at each row can reflect the efficiency of these two designs. Due to the space limit, we combine the results of different games into one table. We will add more clarification to avoid the confusion of comparing the results of different games in the next version.
>
> **The reviewer also asked about BIRD’s effectiveness against different attacks.**
>
> We thank the reviewer for this interesting question. We would like to kindly highlight that we already consider three different attacks in our work, each for one type of game (RL setup): single-agent games, multi-agent competitive games, and multi-agent cooperative games. To the best of our knowledge, there are no other existing backdoor attacks against multi-agent RL. As such, we found a more recent attack against the single-agent RL [1]. This attack considers a similar attack setup as TrojDRL (the one we evaluated) but with a different method of injecting backdoor.
>
> We conduct an extra experiment on the Atari-Breakout and Seaquest game. We train $5$ backdoored agents with this attack and mix them with another $5$ clean agents respectively for each game. We apply BIRD for backdoor detection and removal. The results shown in Table 3 in the submitted pdf file demonstrate the effectiveness of our method against this attack. We will add this experiment to the next version.
>
> [1] Agent Manipulator: Stealthy Strategy Attacks on Deep Reinforcement Learning. Applied Intelligence 2022.
>
> **Finally, the reviewer pointed out that BIRD performs fine-tuning to remove the backdoor, but it does not directly improve the robustness of the RL algorithm itself.**
>
> We thank the reviewer for pointing this out. We will emphasize in the paper that our goal is to robustify a pre-trained agent against backdoor attacks (i.e., identify backdoored agents and remove the backdoor). We do not consider the setup that aims to improve the robustness of the policy training process to reduce the risk of being backdoored.

---

> > ### Author Response · Authors · 2023-08-14
> > **Follow up with the reviewer**
> >
> > Thanks the Reviewer jZ8t again for the insightful comments. Since the discussion phase is about to end, we are writing to kindly ask if the reviewer has any additional comments regarding our response. We are at their disposal for any further questions. In addition, if our response and additional experiments address the reviewer's concern, we would like to kindly ask if the reviewer could reconsider their score.

---

> > ### Comment · Reviewer_jZ8t · 2023-08-21
> >
> > Thank you for your response, which addresses some of my concerns. The additional experiments also add more evidence to showcase the effectiveness. After careful consideration, I decided to keep my score.

---

> > > ### Author Response · Authors · 2023-08-21
> > >
> > > Thank the reviewer for the kindly reply! We are happy that our response could help further demonstrate the effectiveness of our method. We will add the changes to the next version and follow the review's suggestion to improve Table 2.

---

### Official Review · Reviewer_Mdit · 2023-07-07

**Soundness:** 2 fair
**Presentation:** 3 good
**Contribution:** 3 good
**Rating:** 5
**Confidence:** 3

**Summary:**

This paper studies the backdoor defense problem in deep reinforcement learning (DRL) policies. Specifically, the authors proposed the BIRD method to address the limited generalizability and scalability of current practices. By analyzing the unique properties and behaviors of backdoor attacks, the authors formulated trigger restoration as an optimization problem and design a metric to detect backdoored policies.
In addition, a finetuning method is also presented to remove the backdoor, while maintaining the agent’s performance in a clean environment. Experiments are conducted over ten different single-agent or multi-agent environments.

**Strengths:**

- The NeurIPS community finds the topic of backdoor threats in Reinforcement Learning highly relevant.
- The paper is well-written and easy to follow.
- The empirical evaluation seems to be comprehensive.

**Weaknesses:**

- Can you please also report the performances in other popular metrics, e.g., ROC, in addition to the F1 score?
- Can you create some specified baselines tailored to the RL setup? Since I think it might be unfair/inappropriate to apply those defenses for classification models to RL setup.


**Questions:**

Please see my comments above

**Limitations:**

Please see my comments above

---

> ### Author Rebuttal · Authors · 2023-08-07
>
> We thank the reviewer for their positive and constructive review. Please see below for our response and clarifications.
>
> **First, the reviewer asked for an additional ROC curve for the results in Fig. 2.**
>
> Thank the reviewer for the valuable comment. We follow this suggestion and draw the ROC curve in Fig. 1 in the submitted pdf file. We vary the detection threshold $\epsilon$ from -1 to 1 and plot the ROC curve for BIRD in the selected games. Note that the comparison baselines have a fixed detection threshold. Given that their detection performance is low, we do not draw the ROC curves for NC and Pixel. The results are consistent with Fig. 2 in the main text.
>
> **Second, the reviewer asked for an additional baseline method that is suitable for RL setup.**
>
> Thanks for the comment. We totally agree with the reviewer that adding such a baseline can better demonstrate the effectiveness of our method, as the current baselines are not designed for RL. To conduct such a baseline, we can adapt the current baseline to the RL setup. We consider the pixel method [1], which is stronger than NC. To adapt the pixel to RL, we partially leverage the design of our method. That is, we can still model the trigger restoration as solving Eqn.(1) in Lines 135-136. However, instead of modeling the $\Delta$ as a generative process, we can use the pixel objective to model $\Delta$. That is $\Delta = clip(b_p, 0, 1) - clip(b_n, 0, 1)$, where $b_p$ and $b_n$ represents the positive and negative perturbation added to the state. As such, the final objective function for trigger restoration becomes
>
> $\text{max}\sum_s\rho^{\pi}(s) \sum_{a} \pi(s+\Delta) Q_{\pi} (s+\Delta, \pi(s+\Delta)) $, where
>
>
> $\Delta = clip(b_p, 0, 1) - clip(b_n, 0, 1)$.
>
> This objective function can be resolved by the REINFORCE method. We denote this trigger restoration method as pixel-RL. After restoring the trigger, we can use two metrics for this method of detection – the original metric of the pixel and our proposed method denoted as pixel-RL-trigger-size and pixel-RL-reward.
>
> We compare our method with these two adapted baseline under six setups: the two single-agent Atari games: Seaquest and Qbert, with targeted and untargeted attacks, QMIX and You-Shall-Not-Pass in multi-agent games. For each environment, we train 5 clean and 5 backdoored agents. Fig. 2 in the submitted pdf file shows the comparisons of detection results. As we can observe from the figure, BIRD still outperforms both methods (pixel-RL-trigger-size and pixel-RL-reward). We will add this experiment to the next version.
>
> [1] Better Trigger Inversion Optimization in Backdoor Scanning, CVPR 2022.

---

> > ### Author Response · Authors · 2023-08-14
> > **Follow up with the reviewer**
> >
> > Thanks the Reviewer Mdit again for the insightful and positive comments. Since the discussion phase is about to end, we are writing to kindly ask if the reviewer has any additional comments regarding our response. We are at their disposal for any further questions. In addition, if our new experiments address the reviewer's concern, we would like to kindly ask if the reviewer could reconsider their score.

---

> > > ### Comment · Reviewer_Mdit · 2023-08-21
> > >
> > > I appreciate the authors' responses. I have no other questions.

---

> > > > ### Author Response · Authors · 2023-08-21
> > > >
> > > > We thank the reviewer for the kindly reply! We are happy that our response could help address the reviewer's concern. We will add the additional experiment results to the next version of our paper.

---

### Official Review · Reviewer_V9nG · 2023-07-10

**Soundness:** 3 good
**Presentation:** 3 good
**Contribution:** 3 good
**Rating:** 7
**Confidence:** 2

**Summary:**

This paper introduces BIRD (Backdoor Identification and Removal for DRL), a method for detecting and removing triggers in reinforcement learning models. In backdoor attacks, an attacker injects a trigger into the agent's environment during training, leading the agent to take backdoored actions that decrease its actual reward. BIRD addresses the challenge of detecting and removing backdoors from pretrained policies without knowledge of the attack specifications or access to the training process.

BIRD formulates trigger restoration as an optimization problem, identifying the trigger by maximizing the agent's value function. It introduces a novel detection metric based on the actual reward difference before and after adding the restored trigger to the environment. The method effectively detects backdoored agents and employs finetuning with additional regularization terms to remove the backdoor while maintaining performance in the clean environment. Evaluations on various benchmarks demonstrate BIRD's superiority over existing methods, highlighting its generalizability, computational efficiency, and robustness against different attack variations. Overall, BIRD offers an effective solution for detecting and removing triggers in reinforcement learning models, mitigating the vulnerability to backdoor attacks.

**Strengths:**

The paper presents a groundbreaking method for defending against backdoored models in machine learning. The key innovation lies in exploring the total received reward as a means of detection. This unique approach sets it apart from previous works in the field.

The experimental results strongly support the efficacy of the proposed method, showcasing its ability to outperform or at least match the performance of existing techniques. The findings demonstrate that the idea of considering the total reward proves to be highly effective in mitigating the impact of backdoor attacks on machine learning models.

Overall, the paper introduces a novel and promising approach to addressing the backdoor vulnerability in machine learning. The method's success, as evidenced by the experimental results, underscores its potential as a robust defense mechanism against backdoored models.

**Weaknesses:**

However, while the idea of considering the total received reward is novel and promising, there are potential weaknesses that need to be addressed. Firstly, it remains uncertain whether the proposed method's reliance on total reward as a detection mechanism is robust across all scenarios. In particularly noisy or challenging environments, where the model may struggle to learn effectively, it is unclear if the model will consistently receive significantly higher rewards even without the presence of a backdoor.

Additionally, it would be valuable to investigate the potential trade-off between detecting backdoored models and maintaining overall model performance. Since the method focuses on detecting triggers by maximizing the total reward, there is a possibility that it could inadvertently compromise the model's ability to achieve high performance in non-backdoored scenarios. Understanding the impact of this trade-off and ensuring a balance between detection and model performance would strengthen the applicability and practicality of the proposed method.



**Questions:**

Does the main idea work efficiently in different scenarios? More experimental results are needed.

---

> ### Author Rebuttal · Authors · 2023-08-07
>
> We thank the reviewer for their positive and constructive review. Please see below for our response and clarifications.
>
> **First, the reviewer raised concerns regarding the effectiveness of using reward as the detection metric, particularly for challenging games where even a clean agent may struggle to receive a high reward.**
>
> We thank the reviewer for raising this concern. We would like to kindly emphasize that our metric does not rely on the absolute value of the agent’s reward. Instead, we compute the reward difference before and after adding the restored trigger to the environment. To offset the influence of the absolute value, we normalize the reward difference (i.e., $\phi(\pi, \Delta) = (\bar{\eta}(\pi, \Delta) - \bar{\eta}(\pi)) / \eta_{\text{max}}$ in Line 210). A low metric $\phi(\pi, \Delta)$ means that the agent’s performance drops after observing the trigger, indicating the agent may contain the backdoor.
>
> Regarding a challenging game, we can consider the following two scenarios.
>
> (1) We have a weak agent with the ability to receive only very low rewards. Consequently, it is unlikely to be selected as the attacker's target, as the agent is already quite weak. Attackers typically do not have an interest in attacking such weak agents, as they cannot inflict significant damage and would only result in wasted time and effort.
>
> (2) We have a sub-optimal agent that can still perform in the environment but cannot receive a very high reward due to the task's complexity. An attacker will have the incentive to attack such an agent with the goal of significantly reducing its reward. In this case, we will still observe a notable reward drop before and after adding the restored trigger to the environment. We admit that the reward drop will not be that significant compared to a well-trained/near-optimal agent. That is, the metric value will not be very close to -1. But our method can still have the potential to identify the backdoored agent from the clean agent by setting a less aggressive threshold.
>
> To verify this, we conduct an extra experiment using the Atari-SpaceInvaders game. We first prepare $20$ agents. $5$ agents are well-trained clean agents,  $5$ agents are sub-optimal clean agents, $5$ agents are well-trained backdoored agents, $5$ agents are sub-optimal backdoored agents. For all the backdoored agents, we only use the successfully attacked ones. That is, their performance will drop to almost zero after observing the trigger. Otherwise, we treat it as an unsuccessful attack. We mix them together and then apply BIRD to detect the backdoored agents. We vary the detection threshold $\epsilon=-0.5/-0.6/-0.7/-0.8/-0.9$ and report the detection performance. As we can see from Table 1 in the submitted pdf file, as the $\epsilon$ increases, we are able to capture more and more sub-optimal backdoored agents. But we will not include any false positives. We thank the reviewer again for pointing this out. In the next version of the paper, we will clarify that we will not select an overly aggressive detection threshold in case there are some sub-optimal agents that are backdoored.
>
> In addition, we acknowledge that, in general, it is tricky to determine the optimal detection metric. As such, we design our final removal step with the consideration that it will not minimize the effect on the agent’s performance in a clean environment. With that said, conservatively, we can apply the restoration and removal step to all the given agents. In this experiment, as shown in Table 2 in the submitted pdf file, after applying the removal step to all the 20 agents, we can observe that (1) For clean agents, their performance is only marginally affected; (2) For the backdoored agent, their performance in the clean environment keep similar, and their backdoor is removed. This shows that even our detection step fails to capture some backdoored agents. With a conservative solution, the removal step can still remove the backdoor for both optimal and sub-optimal agents. We will emphasize this in the next version.
>
> **Second, the reviewer also raises the concern that whether our method will affect the agent’s performance in a clean environment.**
>
> Thank the reviewer for pointing this out. First, we want to clarify that our restoration and detection step does not vary with the given agent’s policy and thus will not affect its performance. Our removal step indeed may affect the agent’s performance in the clean environment. As mentioned in Section 3.4 Lines 257-264, we add an additional constraint to avoid this in our retraining objective function. As demonstrated in Table 1 in the paper, for all the selected games, BIRD only introduces negligible performance differences in the clean environment after the removal step. Our extra experiment in Table 2 in the submitted pdf file also demonstrates similar efficacy for suboptimal agents and clean agents. We will emphasize this in the next version.

---

> > ### Author Response · Authors · 2023-08-14
> > **Follow up with the reviewer**
> >
> > Thanks the Reviewer V9nG again for the insightful and positive comments. Since the discussion phase is about to end, we are writing to kindly ask if the reviewer has any additional comments regarding our response. We are at their disposal for any further questions.

---

> > > ### Comment · Reviewer_V9nG · 2023-08-15
> > > **Thank you for response my concerns.**
> > >
> > > I have already upgraded my vote to accept.

---

> > > > ### Author Response · Authors · 2023-08-15
> > > >
> > > > We sincerely appreciate the reviewer for updating the score! We will add the rebuttal changes in the next version.

---

### Author Rebuttal · Authors · 2023-08-07

We thank the reviewers for the constructive feedback. We addressed all the comments. Below, we summarize our responses:

We have added all experiments mentioned by reviewers (All the results are in the submitted PDF):

1. We demonstrated the effectiveness of BIRD in detecting poisoned/backdoored agents with a sub-optimal performance by varying the detection threshold (Reviewer V9nG).

2. We demonstrated the effectiveness of BIRD’s backdoor removal for suboptimal agents (Reviewer V9nG).

3. We added the ROC curve for the detection results in Fig.2 in the main text (Reviewer Mdit).

4. We compared BIRD with two additional baselines that tailor existing methods for RL setup. We showed BIRD is still better than these two adaptive baselines (Reviewer Mdit).

5. We tested BIRD against a more recent backdoor attack and demonstrated its effectiveness (Reviewer jZ8t).

6. We demonstrated the BIRD can be adapted to a scenario where the value function is not available (Reviewer M2QE).

7. We showed that BIRD is not that sensitive to the changes in $\epsilon_1$ (Reviewer M2QE).

We have clarified all the questions from reviewers:

**Reviewer V9nG**

1. We clarified that by selecting the proper detection threshold, our method can detect agents with a sub-optimal performance (in complicated games).

2. We demonstrated that our method has a marginal impact on the agent’s performance in the clean environment.

**Reviewer Mdit**

1. We followed the reviewer’s suggestion and added the ROC curves.

2. We followed the reviewer’s suggestion and added two additional baselines that are more suitable for the RL setup.

**Reviewer jZ8t**

1. We followed the reviewer’s suggestion and tested BIRD against another recent attack.

2. We clarified that our method focuses on robustifying a trained agent rather than designing a robust agent training algorithm.

**Reviewer M2QE**

1. We first clarified assuming the availability of value function is a reasonable and practical assumption. We further demonstrate that BIRD is still effective when the value network is not available.

2. We followed the reviewer’s suggestion and added an additional experiment that demonstrates the insensitivity of BIRD to $\epsilon_1$.

3. We clarified our method of identifying and reinitializing neurons during the backdoor removal.

4. We followed the reviewer’s suggestion and update an imprecise description and provide a negative broader impact.

We hope this summary can facilitate the reviewers' evaluation and discussion of our paper. We are at your disposal for any further questions. In addition, We would appreciate it if you could kindly consider updating your scores if our rebuttal has satisfactorily addressed the concerns. Thank you again for your time and consideration.

---

### Comment · Area_Chair_W9xJ · 2023-08-10
**Author-Reviewer Discussion phase (Aug 10-16)**

Today begins the Author-Reviewer Discussion phase, which lasts 1 week (**Aug 10-16**).

I ask the reviewers to please **carefully read all other reviews and the author responses
and (if appropriate) respond to author responses promptly.**   If you've read the author response, please take the time to leave a comment, even if you have nothing to add.

I also encourage both authors and reviewers to monitor OpenReview for further comments in order to enable as much back-and-forth as possible during this short period.

**Regarding this submission:**
Reviewer M2QE gave a confident negative review, criticizing the assumption that the value function will be available.  The other reviewers all voted for acceptance, although two reviewers find it borderline, and also suggested improvements.

It seems to me that the authors have responded in detail to all of the reviews, and I ask the reviewers to consider whether these responses have effectively addressed their concerns.  Also, all reviewers should consider whether Reviewer M2QE's criticism affects their evaluation of the work or raises further questions they wish to ask the authors.

---

> ### Author Response · Authors · 2023-08-10
>
> Thank you for initiating the Author-Reviewer Discussion phase for our submission. We value this opportunity to interact with reviewers and address their feedback.
>
> We have carefully reviewed all the reviews and comments, including Reviewer M2QE's insightful critique on the value function assumption. In our response, we first provided practical scenarios validating our assumption about value function, then we proposed an alternative approach for cases where the value function isn't available.
>
> We have also addressed all of the suggestions and concerns from four reviewers, aiming to improve the quality of our work based on their valuable feedback.
>
> During this Author-Reviewer Discussion phase, we will actively participate on OpenReview, and we look forward to any further questions or discussions that the reviewers may have. We understand the importance of this back-and-forth dialogue and appreciate the opportunity to provide additional explanations as needed.
>
> Thank you for your ongoing support during this review process.

---

### Decision · Program_Chairs · 2023-09-21

**Decision:**

Accept (poster)

**Comment:**

This paper proposes a novel method, BIRD, which allows for both identifying and removing backdoors from Deep RL policies.  The authors were able to effectively address reviewers’ concerns during the rebuttal phase, and performed the additional experiments suggested.  A primary novelty is that the algorithm only requires access to a clean environment, however, reviewer M2QE was originally concerned that the method required access to the value function as well.  However, the authors demonstrated that BIRD can be successfully applied even without such access.  After much discussion, and improvements, reviewers all recommend accepting this paper, and I am happy to recommend acceptance as well.